# Potential Hotspots of Hamadryas Baboon–Human Conflict in Al-Baha Region, Saudi Arabia

**Ghanem Al-Ghamdi [1,*], Abdulaziz Alzahrani [2], Saleh Al-Ghamdi [1], Salihah Alghamdi [3], Abdullah Al-Ghamdi [4], Wael Alzahrani [4] and Dietmar Zinner [5,6,7]**

1    Departments of Biology, College of Science, Al-Baha University, Al-Baha 65799, Saudi Arabia;
     sb.alghamdi@bu.edu.sa
2    Department of Architecture, College of Engineering, Al-Baha University, Al-Baha 65799, Saudi Arabia;
     azahran@bu.edu.sa
3    Department of Mathematics, College of Science, Al-Baha University, Al-Baha 65799, Saudi Arabia;
     alghamdi.ss@bu.edu.sa
4    Department of Biology, College of Arts and Science Baljorashi, Al-Baha University,
     Al-Baha 65799, Saudi Arabia; ab.showail@gmail.com (A.A.-G.); wail1_2008@hotmail.com (W.A.)
5    Cognitive Ethology Laboratory, German Primate Center, Leibniz Institute for Primate Research,
     37077 Göttingen, Germany; dzinner@gwdg.de
6    Department of Primate Cognition, Georg-August-University of Göttingen, 37077 Göttingen, Germany
7    Leibniz Science Campus Primate Cognition, 37077 Göttingen, Germany
*    Correspondence: galghamdi@bu.edu.sa

**Abstract:** In recent decades, conflicts between hamadryas baboons (*Papio hamadryas*) and the rapidly growing human population in the mountainous areas of Western, Southwestern, and Southern Saudi Arabia have accelerated. This conflict, historically occurring mainly between farmers and baboons, has now moved to the urbanized areas in the baboon range and is mainly caused by the common use of spatial and other resources by baboons and humans. The goal of this study was to describe the spatial distribution of baboon groups and to estimate the population size of baboons in the Al-Baha region. The results indicate that baboons are present in all administrative areas of the Al-Baha region with a concentration along the mountain chain running from northwest to southeast and the western part of the region. As expected, rubbish dumps constitute baboon hotspots due to the large amounts of human-derived food. However, the baboons also travel into towns for foraging. The prevention of baboon accessibility to human-derived food would be an important step to reduce causes of conflicts between humans and baboons.

**Keywords:** *Papio hamadryas*; urban area; human-baboon conflict; Saudi Arabia

## 1. Introduction

Hamadryas baboons (*Papio hamadryas*) are the only non-human primates occurring in the Arabian Peninsula [1,2]. Their distribution is confined to the southwestern part of the peninsula from Al Akhal (Saudi Arabia) in the north to Aden (Yemen) in the south [3–5]. Over the last 50 years, Saudi Arabia has seen an extensive and fast economic development. The country's human population rose from 7,009,466 in 1974 to 34,110,821 in 2021 https://www.stats.gov.sa (accessed on 1 May 2021) with a high degree of urbanization, considerably so in areas within the hamadryas baboon range. Some baboon populations have adapted to the development in the region and are now exploiting human-derived food sources around these urban centres (e.g., Abha and Al-Baha) [4]. In addition to the conversion of natural habitats into human settlements and business areas, this development has caused remarkable extensions of cities and an increase in the purchasing power of the human population, subsequently resulting into a rise in waste disposal problems, which is largely managed by the establishment of open waste-management facilities [6,7]. Such facilities provide "high quality" food to baboons and other "scavengers", e.g., feral dogs. In addition,

local people and tourists distribute increasing amounts of food to baboons and feeding baboons has become an attraction for many people. Due to the access to human-derived food and the extirpation of natural predators, such as the Arabian leopard (*Panthera pardus nimr*), the number and size of local baboon populations have increased in some areas causing destruction and severe hygienic and medical problems [8].

In the 1980s, it was reported that at many sites between Taif and Abha, except for the naturally rich Wadi Hesswa, the baboons gathered about half of their food from human sources, mainly from rubbish dumps [3]. The dumps at Al Hada and Abha were so rich that the baboons spent many hours throughout the day resting near them and only travelled short distances [9]. Many adult males at Abha were overfed and showed signs of obesity [3]. In addition, the destruction of the natural vegetation within large parts of the baboon range by overgrazing and conversion to urban constructions might also have forced the baboons to shift to human-derived food [7].

Historically, hamadryas baboons might have caused problems for farmers due to crop raiding, but the large baboon populations now live near and in cities, causing different problems [10]. Biquand et al. (1989) classified the hamadryas baboons of Saudi Arabia according to these problems into four levels: (1) feeding on natural vegetation in relatively low population densities; (2) raiding crops or gardens more or less frequently and feeding only partially on natural vegetation; (3) scavenging at rubbish dumps; and (4) consuming food directly provided by humans and feeding very little on natural vegetation. Scavenging groups can become very large, including as many as 400 individuals (unpublished data). Baboons also wait near roads for food thrown at them from cars, invade gardens and houses, and visit public parks where people feed and tease them for entertainment (own observation, Figure 1).

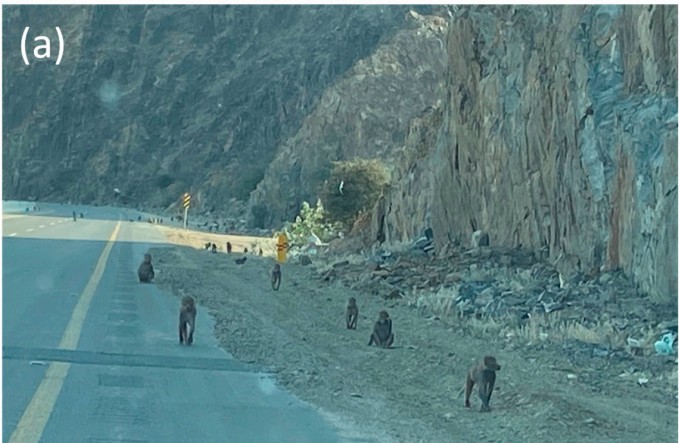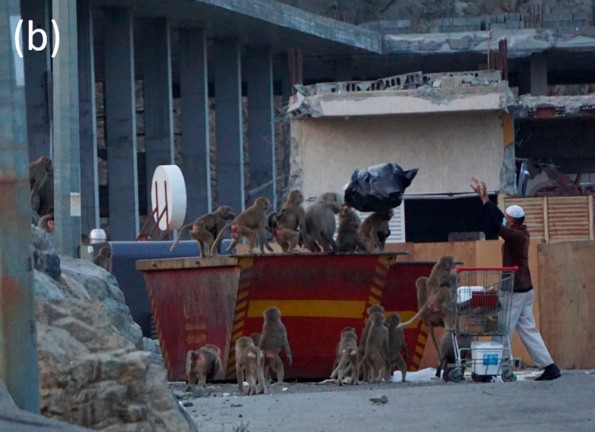

**Figure 1.** (**a**) Hamadryas baboons travelling along the roadside near Al-Baha looking for food provided by humans. (**b**) Hamadryas baboons have easy access to high-quality food at a garbage container at the Mall/Raghdan Park (MRP) area due to inappropriate human actions.

The conflicts between baboons and humans create unpleasant conditions in which baboons can become not only a nuisance but can be dangerous. They might attack people to steal food or to defend their infants. Due to the increasing contacts between baboons and humans, baboons also constitute a potential problem for public health. They are a reservoir for a significant number of zoonotic pathogens. Zoonotic intestinal parasites such as *Giardia* sp., *Ascaris* sp., *Entamoeba histolytica*, *Balantidium coli*, *Enterobius* sp., *Trichuris* sp., Hookworm, *Hymenolepis nana*, and *Schistosoma mansom* were described in the south-western, western, and northern regions of Saudi Arabia [11–13]. Several bacterial species were detected as well including *Staphylococcus aureus*, *Campylobacter* sp., *Clostridium* sp., *Mycobacterium* sp., *Shigella* sp., and *Salmonella* sp. [14]. Finally, evidence of viral infection such as rabies and coronavirus has been described from baboons [15,16].

Within the Al-Baha region, in the southwestern part of Saudi Arabia, with a human population of more than 400,000, the hamadryas baboon population is larger than in other areas [4]. As in other parts of their Arabian range, the baboons in Al-Baha use rubbish dumps, dustbins, and waste containers of supermarkets, public parks, and private gardens within city centres and in their surroundings. While searching for food and manipulating objects, the baboons destroy infrastructure such as phone lines, power lines, water pipes, private properties, and local businesses. In addition, they contaminate areas, including children's playgrounds with skin contaminants, faeces, and bodily fluid, and the number of human injuries caused by baboons has increased considerably within the last few years (unpublished data). This is of particular concern because large packs of feral dogs live in sympatry with the baboons at the rubbish dumps and other baboon locations, which might increase the probability of zoonoses like rabies from dogs to baboons and humans [15,17].

The baboon situation in Al-Baha has become more severe and the potential for human–baboon conflict increased within the last three decades. As a first step to contribute to the mitigation of the problems, we did a field survey aiming to identify the major ranging areas of the baboons and the localities of frequent conflict among baboons and humans, to estimate baboon population size, and to provide information on how the number of baboons changes at selected sites during the course of the day. The outcome of this work will provide local authorities with reliable data on the magnitude of the problem.

## 2. Materials and Methods

### 2.1. Study Area

The Al-Baha region is located in the southwestern mountains of Saudi Arabia, about 300 km southeast of Jeddah (20°00′45″ N 41°27′55″ E). The total area of the Al-Baha is approximately 11,000 km$^2$ (HABITAT, 2019). The Al-Baha region consists of two major sectors: a lowland coastal plain in the west, known as "Tihama", and a mountainous region with an altitude of 1500 to 2450 m in the east, known as "Al-Sarat or Al-Sarah". The mountain area is a part of the Al-Sarawat Mountains [18]. Al-Baha is thus a transition zone between the arid coastal plain, west-facing rocky mountains, and east-facing semi-desert mountains. It consists of a diverse range of habitats, including woodland, shrubland, and grassland [19]. In these areas, we collected data on the geographic positions of baboon sightings, the number of baboons at the sites, as well as data on their temporal appearance at selected sites where the contact between baboons and humans was assumed to be frequent. The study period lasted between September and October 2021 and thus covered only the limited inter-seasonal period.

### 2.2. Baboon Occurrence

We visited potential baboon sites based on information available from local authorities (e.g., if residents complained about baboons), and social media (e.g., if people posted pictures when encountering baboons or feeding baboons). We determined the geographic coordinates of all baboon positions where we encountered baboon groups with a handheld global positioning system (GPS) device and for each baboon sighting, we added the observed number of baboons. Around local accumulations of occurrence points, we created polygons representing areas inhabited by baboons. These polygons are identical to the home ranges of the local baboon troops. However, since hamadryas baboons have overlapping home ranges [20,21], these polygons often represent partly overlapping home ranges of more than one troop. In cases where occurrence data did not show an accumulation pattern, we recorded just "single observations".

We annotated these data onto topographical maps in ArcGIS 10.8. To account for the mobility of the baboons, we added a buffer with a radius of 15 km to each sighting location. The buffer size is consistent with the maximum recorded daily travel path length for baboons [21,22].

### 2.3. Temporal Changes in Baboon Numbers

At four arbitrarily chosen sites in the Al-Baha region, we collected detailed data on the temporal sequence at which baboons appeared and disappeared at the site during the course of the day. The question here was: at what times of the day was the number of baboon highest at each site. The four sites were (1) the Mall/Raghdan Park (MRP) where baboons occupy the park and have access to the local commercial centre; Prince Sultan Park (PSP) including neighbouring villages and small residential and farming areas; (3) the waste-management facility (WMF) area of Al-Baha city; and (4) Al-Baha University Campus (BUC), an area densely covered by student and faculty housing.

The number of baboons at each site was monitored by three well-trained observers for three consecutive days in the period from September to October 2021. The sites were scanned for baboons from 5 a.m. to 7 p.m., except for the waste-management facility (WMF) where the access time for observers was restricted by the administration. Essentially, a snapshot of the number of baboons was performed at the top of the hour every hour between 5 a.m. and 7 p.m. We took videos and photos of the baboons which we used to control our on-site estimations of baboon numbers.

### 2.4. Statistics

Since we regard our study as a first preliminary approach and we therefore were not able to collect sufficient data for proper statistical testing, we only used descriptive statistics to present our findings.

## 3. Results

### 3.1. Occurrence of Hamadryas Baboons

During our survey, we found hamadryas baboons in all districts of the Al-Baha region. These locations included forests, regional parks, developed villages, school zones, farms, and the city centre. The distribution of baboon sites was mainly concentrated along the mountain region that dissects the Al-Baha region from northwest to southeast (Area 1, Figure 2). This part of the region is also the preferred area for human settlement (https://www.stats.gov.sa). The second area where occurrence points accumulated lies in the western parts of the Al-Baha region (Area 2, Figure 2). This area is heavily populated by humans too and includes farms, commercial activities, and local recreation parks. The Al-Aqeeq district (Area 3, Figure 2) is the largest district of the Al-Baha region but with the lowest human population density. Similarly, the number of baboon sites in this district was also low.

With a maximum daily travel distance of nearly 14 km, the areas of possible conflicts between humans and baboons are depicted in Figure 3. It became obvious that the main areas of potential conflict are Al-Baha City, Bani Hassan and Al-Mandaq. West of this central region, the governorate of Qilwah seems also to have a high potential for conflict.

### 3.2. Baboon Population Size

We categorized the occurrence sites according to their estimated local population size into five classes. Sites with larger and smaller local populations were found in all three main study areas (Figure 4).

At the four sites, the baboons were counted for three consecutive days in each site. The largest number of baboons was found at the waste-management facility (WMF, Figure 5) area with more than 2500 baboons, followed by the Mall/Raghdan Park (MRP, more than 250 baboons), 133 baboons at Al-Baha University Campus (BUC), and 65 baboons at Prince Sultan Park (PSP).

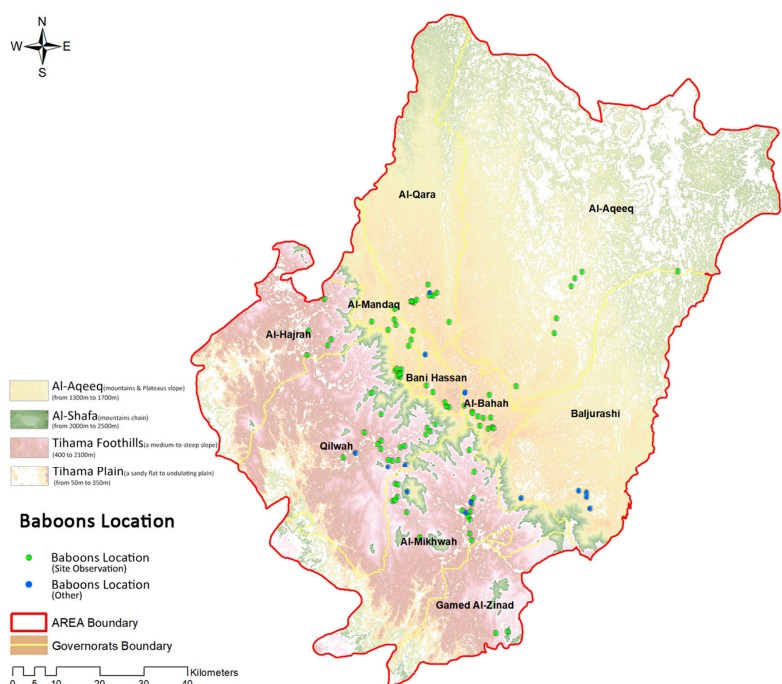

**Figure 2.** Distribution of hamadryas baboon sites (green dots) within the Al-Baha region. The occurrence of baboons concentrates along the mountain chain running from northwest to southeast (Area 1) and the western part of the region (Area 2). In contrast, the density of occurrence points of baboons in the coastal lowlands (Tihama Plains) and the large eastern Al-Aqeeq district (Area 3) is comparatively low. The Tihama Plain is sandy, flat to undulating, and the Tihama foothills comprise a medium-to-steep slope with valley gullies descending westwards. Northeast of the foothills follows steep slopes which rise further to an undulating high-altitude plateau. Northeast of the plateau, the landscape drops again into the Al-Aqeeq region. Most likely, the main areas of potential conflict are Al-Baha City, Bani Hassan, and Al-Mandaq.

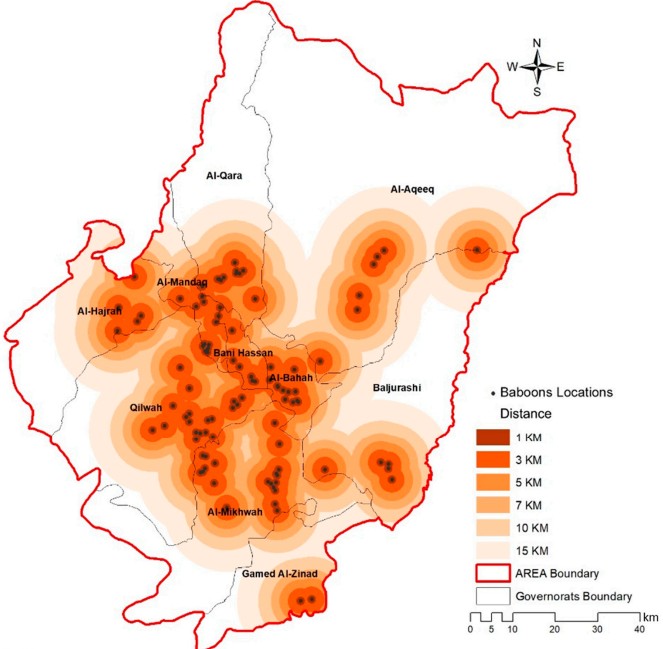

**Figure 3.** The spatial distribution of baboon occurrence points and corresponding buffer areas indicating the potential ranging area of the baboon. The maximum daily travel distances of baboons are 14 km [21,22] and we regard these buffer areas as areas of potential human–baboon conflict.

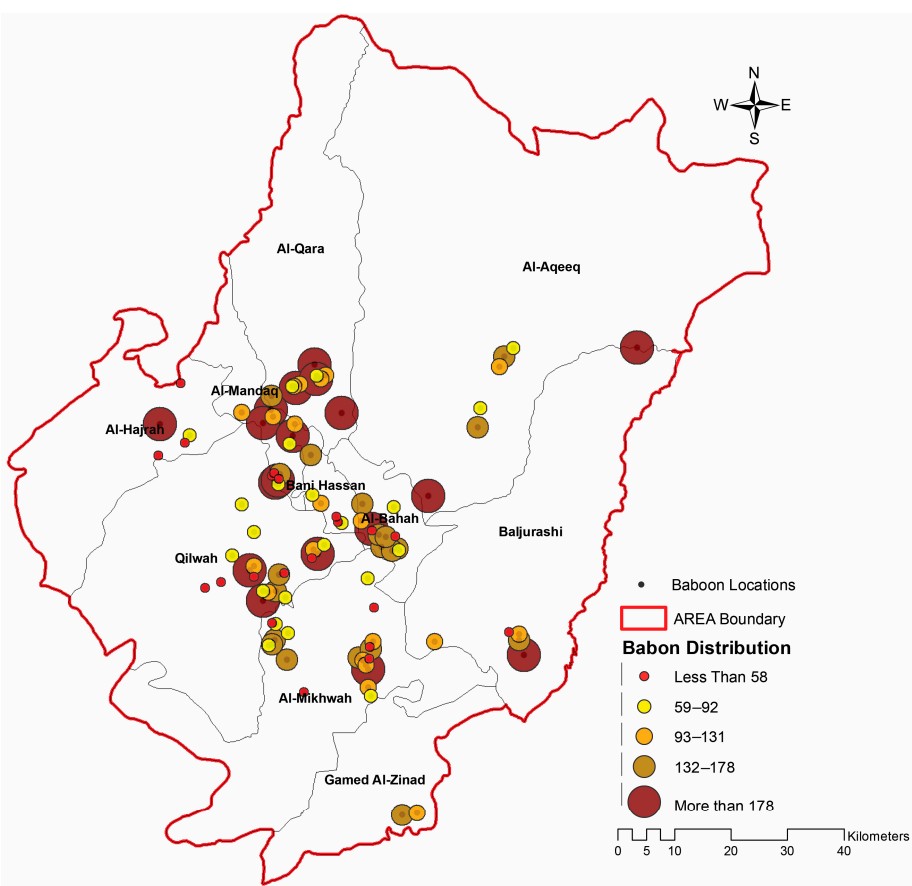

**Figure 4.** Estimated baboon population sizes at baboon occurrence points in the Al-Baha region.

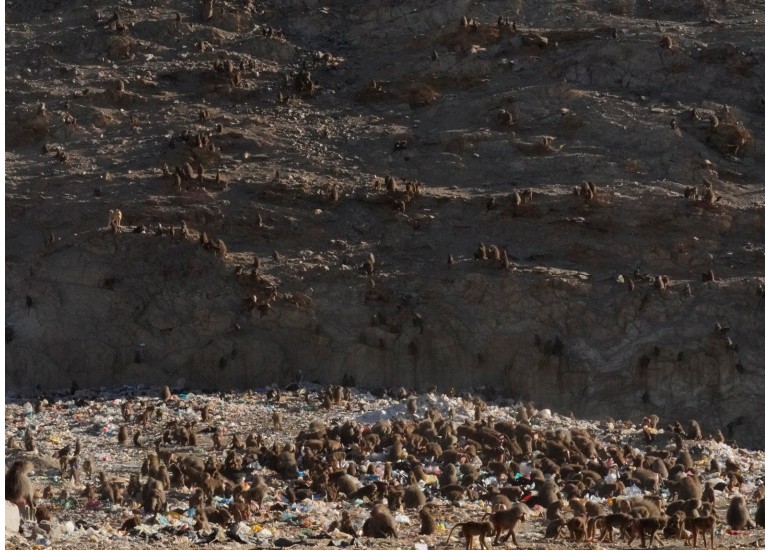

**Figure 5.** A large number of hamadryas baboons at the waste-management facility (WMF) where the baboons wait every day for unloaded food.

The number of baboons did not remain stable over the course of the day. When we counted the baboons every hour, the following pattern occurred (Figure 6). For three out of four sites, most of the baboons were seen at the sites in the early morning and late afternoon. For the fourth site, the waste-management facility (WMF), we were not able to count the baboons in the afternoon due to administrative restrictions.

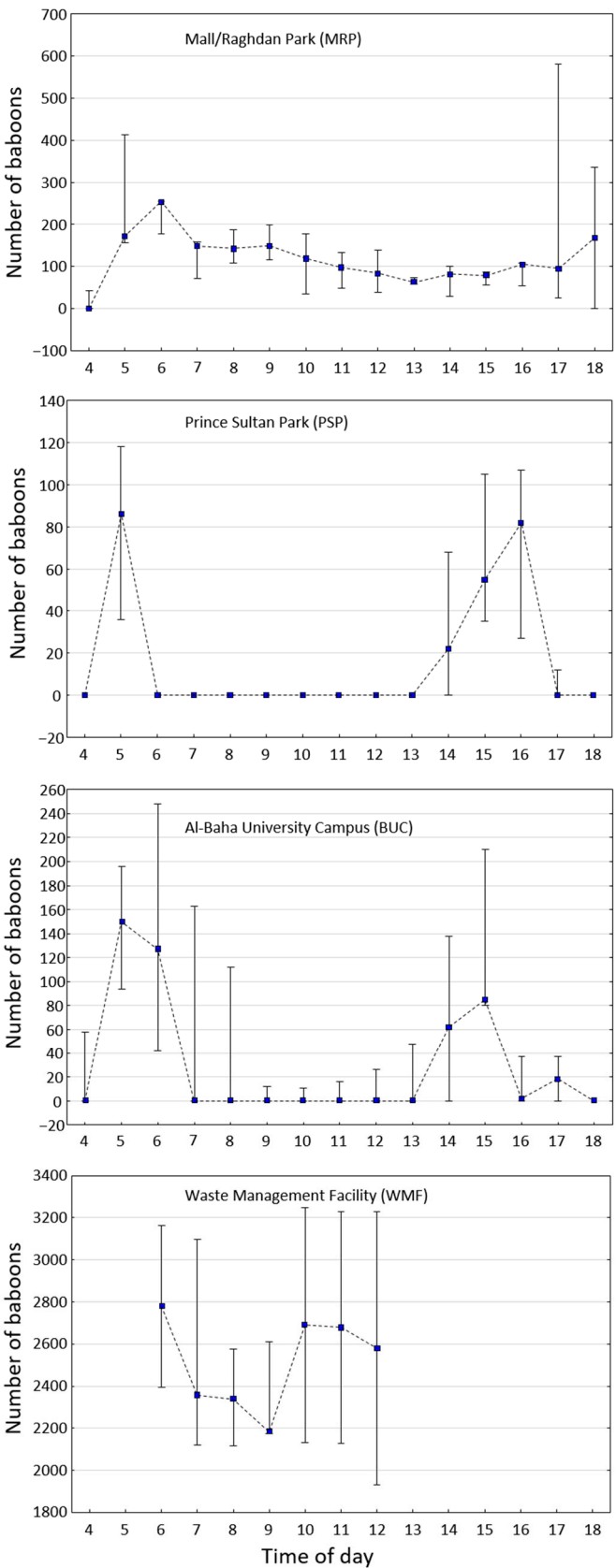

**Figure 6.** Median (min–max) number of hamadryas baboons during the course of a day at four study sites. The numbers of baboons have been counted on three consecutive days at each site.

## 4. Discussion

Early reports described a limited number of baboons in the southwestern region of Saudi Arabia, hardly surviving in cliffs and near nomads [1]. However, their population increased over the years and in the 1990s their numbers in the Al-Baha region exceeded population estimates in the Abha and Taif regions and rose further [4]. Detailed information on the distribution and population size within the area, however, remained largely unknown [4,6]. Our data indicated that all districts of the Al-Baha region are inhabited by baboons extending from the eastern (Al-Aqeeq) to the western (Al-Makhwah) and northern (Al-Qara) to the southern (Baljorashi) districts. Nonetheless, variation exists in the number of hotspots of baboon occurrence among various areas of the Al-Baha region with a higher number of spots in the high-altitude mountains (Al-Shifa) followed by the Tihama foothills where more people live. Due to the habitat characteristics, feeding sites as well as sleeping sites are abundant here. In the meantime, the data also suggest that baboon population densities are higher in these two areas. An exceptional location that had the highest density of baboons was the waste-management facility (WMF) where they obtain an abundance of food that is human-derived food without having to travel long distances, as they intelligently utilize the aforementioned collection areas of waste management. Also, the density in the central mountains and western region was large, as well where baboons have access to various human-derived food sources including waste bins at supermarkets and in recreation parks (tourist sites), gardens, houses, and small farms. In addition, local people as well as tourists tend to manually feed baboons on purpose either for pleasure or seeking good deeds. Finally, baboons are grouping and regrouping in such locations during the travelling searching for food crossing the distances that separate sleeping sites. Therefore, the identification of hotspots for a specific group may require further work of monitoring.

The activity of the baboons and their daily routine was most likely influenced by a combination of food availability and human behaviour. This relationship became particularly obvious at the four locations, where we performed detailed observations; the Mall/Raghdan Park (MRP), Prince Sultan Park (PSP), and Al-Baha University Campus (BUC), as well as the waste-management facility (WMF) area. In these locations, the number of baboons peaked at times when food availability was high or human activity was relatively low, i.e., early morning and late afternoon. At the end of the day, the baboons returned to their sleeping cliffs in the nearby mountain. With the exception of the waste-management facility (WMF), large parts of the foraging areas of the baboons are shared with people carrying out various professional and recreational activities. Therefore, these areas are the places where conflicts between baboons and humans can arise. Baboon attacks of people have been reported on the local news and social media in which several cases were admitted to hospitals (unpublished data). In addition to the physical attacks, the possibility of transmission of diseases has to be considered. These locations are heavily visited and used by people and the risk of disease transmission is high. Several reports have described transmissible pathogens in baboons [11–16]. Therefore, the continuous monitoring of disease epidemics cannot be overstressed.

Since the 1990s, various methods have been tested to mitigate the baboon–human problem in Saudi Arabia [6]. Since systematic culling of baboons is ethically unacceptable, in the 1990s, a series of measurements were tested to deter the baboons from entering towns and cities or reducing their numbers [6,9]. These measurements included fertility control by chemical vasectomy of males, a relatively elaborate and expensive method that yielded only short-term effects or was tested only on captive baboons [6]. Compared to other methods of male sterilization such as surgical vasectomy, this procedure can be done quite rapidly under field conditions (one minute per animal) at relatively low costs and with a low risk of haemorrhage and infection. The closure of open rubbish dumps and other restrictions of the accessibility of baboons to human-derived food (waste bins, feeding of baboons) probably may have a longer-lasting effect.

**Author Contributions:** G.A.-G., A.A. and D.Z.: Conceptualization, methodology, formal analysis, writing—original draft preparation, G.A.-G.: funding acquisition, A.A.: software, A.A.-G. and W.A.: methodology, S.A.-G. and S.A.: writing—review and editing. All authors have read and agreed to the published version of the manuscript.

**Funding:** The authors extend their appreciation to the Deputyship for Research and Innovation, Ministry of Education, Saudi Arabia, for funding this research work through the project number MOE-BU-9-2020.

**Institutional Review Board Statement:** The study was conducted in accordance with the Declaration of Helsinki, and approved by the Ethics Committee of Al-Baha University (protocol code 43104478), 2021.

**Data Availability Statement:** All raw data can be provided by the corresponding author upon reasonable request.

**Conflicts of Interest:** The authors declare no conflict of interest.

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
