# Peer review of "Potential Hotspots of Hamadryas Baboon–Human Conflict in Al-Baha Region, Saudi Arabia"

_diversity, doi:10.3390/d15111107_

Round 1
Reviewer 1 Report
Dear Authors,
Thank you for your effort on this document. I have some suggestions for you to help you improve your manuscript.
Line 74. "Baboons also constitute a potential problem for public health." Please consider "increasing baboons in contact with people who might develop a public health problem".
Line 90. "In addition, they contaminate areas." Please be more specific about the term "contaminate".
Line 100, "to estimate baboon population size and describe their daily activity routines". This needs to be described in the methods and results. Please consider to rewrite this paragraph to make clear what you did.
Line 101. Please be more specific about how your work will provide local authorities to reduce the area conflicts.
Line 118. In your methods, you need to mention how long the study is. You only say three days, but for how long (months)? This will give a better idea about the strength of the study. Also, to compare the number of records in each place.
Line 121. Could the geographic coordinates of all baboons could be related to the changes during seasons? And also, this could explain the overlapping patterns in home ranges.
Line 141. Three days for how long?
Line 144. Your manuscript mentions that you will describe the daily spatial and temporal movement pattern, but the methods need to clarify this. Do the sites have the same time for data collection? It will be advantageous to have a more detailed description.
Line 163. You mention a "comparative low," but no statistical data supports this statement.
Line 169. Although the possible conflicts between humans and baboons are related to the proximity, I consider not mentioning"obvious" perhaps is better to remove this word.
206. You mention that the Al-Baha region had a significantly higher number of spots in the high-altitude mountains. Still, it would be best to work with statistical analysis to do this affirmation in your study.
208. You mention that the sleeping sites are more abundant, but compare to which areas?
209. Data suggest that baboon population densities are higher in these two areas, but no statistical analysis was run.
221. You argue that food availability and human behaviour most likely influenced baboon activity and daily routine. However, it will be interesting to see data to support this affirmation.
226. How did you measure the food availability or human activity to discuss this?
Please consider being more detailed in statistics and describe the variables. In that way, your study will have more support.
I hope these observations can help to improve your manuscript.
Author Response
Dear
We would like to thank you for your comments
Please find the attached letter.
Regards

Reviewer 2 Report
see attached file.

Author Response
Dear,
We would like to thank you for your comments.
Please find the attached letter.
Regards

Round 2
Reviewer 1 Report
Dear Authors,
Your paper improved, and I am happy with your changes.
Sincerely,